# MRD Monitoring by Multiparametric Flow Cytometry in AML: Is It Time to Incorporate Immune Parameters?

**DOI:** 10.3390/cancers14174294

**Published:** 2022-09-01

**Authors:** Ilias Pessach, Theodoros Spyropoulos, Eleftheria Lamprianidou, Ioannis Kotsianidis

**Affiliations:** 1Department of Hematology, Athens Medical Center, 11634 Athens, Greece; 2Department of Hematology, University Hospital of Alexandroupolis, Democritus University of Thrace, 69100 Alexandroupolis, Greece

**Keywords:** acute myeloid leukemia (AML), multiparametric flow cytometry (MFC), measurable residual disease (MRD)

## Abstract

**Simple Summary:**

Measurable residual disease (MRD) is emerging as an important prognostic and predictive biomarker in acute myeloid leukemia (AML). However, its use is currently hampered by the disparity and lack of harmonization between the available MRD methodologies. In addition, the current assessment of MRD in AML focuses only on the quantification of the residual leukemic burden, without addressing the parallel alterations of the antineoplastic immune response that can critically affect the course and outcome of AML, often despite MRD persistence. Incorporating parameters of immune competence provides more consistency with the biological concept of MRD and may lead to higher accuracy. Multiparameter flow cytometry (MFC) is a highly efficacious and sensitive technology for the thorough and synchronous investigation of the kinetics of both antitumor immunity and the leukemic clone. MFC-based MRD provides the platform for the development of a composite leukemia- and immune-based biomarker which can outcompete the current MRD assessment.

**Abstract:**

Acute myeloid leukemia (AML) is a heterogeneous group of clonal myeloid disorders characterized by intrinsic molecular variability. Pretreatment cytogenetic and mutational profiles only partially inform prognosis in AML, whereas relapse is driven by residual leukemic clones and mere morphological evaluation is insensitive for relapse prediction. Measurable residual disease (MRD), an independent post-diagnostic prognosticator, has recently been introduced by the European Leukemia Net as a new outcome definition. However, MRD techniques are not yet standardized, thus precluding its use as a surrogate endpoint for survival in clinical trials and MRD-guided strategies in real-life clinical practice. AML resistance and relapse involve a complex interplay between clonal and immune cells, which facilitates the evasion of the leukemic clone and which is not taken into account when merely quantifying the residual leukemia. Multiparameter flow cytometry (MFC) offers the possibility of capturing an overall picture of the above interactions at the single cell level and can simultaneously assess the competence of anticancer immune response and the levels of residual clonal cells. In this review, we focus on the current status of MFC-based MRD in diverse AML treatment settings and introduce a novel perspective of combined immune and leukemia cell profiling for MRD assessment in AML.

## 1. Introduction

What is actually “remission” in AML? Acute myeloid leukemia (AML) is a group of heterogeneous clonal myeloid disorders encompassing a wide range of molecular alterations, as reflected by the plethora of disease subtypes in the current WHO classification [1]. AML particularly impacts older adults conveying a poor prognosis and only 35–40% of patients under 60 years of age and 5–15% of those older than 60 years of age will attain long term remissions [1]. Intensive chemotherapy has been the standard tool to fight AML for many decades, and in several cases the cytotoxic therapy is followed by allogeneic stem cell transplantation (alloSCT) [2]. A wide range of pretreatment cytogenetic and mutational profiles informs prognosis in AML and allows for categorization of patients with adverse, intermediate, or favorable-risk disease as outlined by the European Leukemia Net (ELN) criteria [3]. However, the current prognostic assessment of AML cannot precisely predict the curative potential of chemotherapy alone or accurately estimate the benefit of alloSCT strategies. In addition, the advent of targeted therapies has, in several cases, disconnected the quality of response from survival benefit, thus adding another layer of complexity to the prognostic assessment of AML [4,5]. According to the latest ELN criteria [6] an AML patient is considered to be in complete remission (CR) if all of the above criteria are fulfilled: bone marrow blasts <5%; absence of circulating blasts or blasts with Auer rods; absence of extramedullary disease; ANC ≥1.0 × 10^9^/L (1000/μL); and platelet count ≥100 × 10^9^/L (100,000/μL).

Complete remission (CR) achievement is a necessity for the cure and prolonged survival of AML patients. However, as morphological remission has long been known to be insensitive for relapse prediction and approximately 50% patients who achieve CR eventually relapse, the notion of “measurable residual disease” negativity (MRD-negative CR) has recently been introduced by the ELN as a new outcome definition [7,8]. MRD indicates the presence of leukemia cells at frequencies below that of routine measurement, either by morphology or cytogenetics, with sensitivity down to one in 10^−4^ to one in 10^−6^ of total leukocytes compared to one in 20 in standard morphology [9]. The added prognostic value of MRD in patients in morphologic CR is well established [8,9,10,11]; however, the lack of standardization and comparability among the various methodologies and the reporting of results currently precludes, in most circumstances, leveraging MRD-guided strategies (Table 1).

## 2. Multiparametric Flow Cytometry-MRD (MFC-MRD) Testing in AML

The investigation and use of cell surface antigens to differentiate normal hematopoietic cells from leukemic blasts started decades ago. Baker et al., in 1974, introduced a rabbit-derived antiserum that was non-reactive with normal cells but highly reactive with leukemic blasts from patients with ALL and AML, presenting evidence that antigenic differences existed between leukemic and non-leukemic cells [16]. Greaves et al. first proposed that the use of immunofluorescent reagents and antibodies directed against leukemia-associated antigens could detect residual leukemic cells, helping to monitor patients for early signs of relapse [17].

The main advantages of MFC-MRD in AML, namely the high applicability (>90%), short turnaround time, relatively high sensitivity, and the possibility to discriminate between living and dead cells [7,8,12,18,19], make MFC-MRD the ideal method to target real-time therapeutic clinical decision making [9,13,14,15]. The sensitivity of the MFC-based MRD assessment is 10^−3^ to 10^−5^ depending on the method, the number of cells analyzed, the design of the panel, the instrument settings, the number and type of antibodies used, and a clinically validated cut-off point for MRD positivity [7,8,11]. The predictive value of MFC-MRD assessment in patients with AML has been studied in several settings, but most data are currently derived from studies with intensive cytotoxic regimens and in settings with allogeneic stem cell transplantation, whereas data on low-intensity therapies are still limited [11] (Table 2).

### 2.1. MFC-MRD after Intensive Chemotherapy

The pediatric protocol AML02 investigated the use of targeted chemotherapy and alloSCT, and a better clinical outcome was reported compared to previous studies. In this study, risk categories were based on the genetic/cytogenetic profile of the patients and MRD findings. FCM-MRD was applied after the first cycle of chemotherapy and treatment was intensified by gemtuzumabozogamicin in patients with high levels of MRD [34]. In the international prospective pediatric study conducted by Langebrake et al., 2006 [21], it was demonstrated that the detection of residual blast cells by flow cytometry at early follow-ups (until day 84) was a significant predictor of treatment outcome, defined as 3-year EFS. The authors utilized MFC-MRD at four timepoints, namely 15 days from start of therapy, before the second induction and before the first and second consolidation course. Already in the early course of therapy before the start of the second induction, at day 28 from diagnosis, MFC-MRD exhibited high prediction accuracy. Coustan-Smith et al. introduced the analysis of residual blasts at the end of remission induction therapy using a specific panel of markers/combinations based on the initial immunophenotype. They established and validated a method to monitor residual disease in children with AML, and introduced immunophenotypes suitable for the detection of residual disease that can be identified by four-color flow cytometry, which allows for sensitivity of detection to be as much as 100 times greater compared to morphological examination [25]. Flow cytometric evidence of leukemia after the initiation of therapy seems to be an independent prognostic factor associated with poor outcome in pediatric studies [20].

Regarding adult patients, in the study by Liu et al. [22], MFC-MRD status after two consolidation cycles had greater accuracy in predicting relapse compared to MRD status after induction. This result was consistent for all different sub-groups (young vs. elderly patients, ELN cytogenetic low- or intermediate-risk patients). By contrast, Minetto et al., 2021 [37], reported that an earlier MRD evaluation timepoint may provide the most significant information on outcome in cases of induction regimens with the combination of fludarabine plus high dose cytarabine. MRD monitoring indicated that patients who had detectable leukemia after induction but achieved MRD negativity after the second consolidation had the same prognosis as those with a negative MRD at both timepoints. Similarly, in the HOVON/SAKK (Dutch–Belgian Haemato-Oncology Cooperative Group and the Swiss Group for Clinical Cancer Research) AML42a study, a prospective analysis of MRD using Leukemia-Associated Immunophenotypes (LAIPs), MRD positivity after cycle 2 was associated with a higher risk of relapse, with 4-year relapse-free survival (RFS) of 23% and relapse incidence of 72% compared to 52% and 42%, respectively, for MRD-negative patients [26].

The QUAZAR AML-001 Maintenance Trial is the first large prospective, double-blind, placebo-controlled randomized trial with long-term longitudinal assessment of MRD in patients with AML in remission. In this trial, administration of the oral formulation of azacytidine (CC-486) as maintenance was associated with significantly longer overall and relapse-free survival than placebo among older patients with AML who were in remission after chemotherapy. In both treatment arms, MRD-positive status (≥0.1%) after induction ± consolidation was associated with significantly shorter OS and RFS compared to MRD-negative status [38].

### 2.2. MFC-MRD after Lower Intensity Treatment

The prognostic value of MRD assessment after intensive chemotherapy and allogeneic hematopoietic stem cell transplantation (SCT) has been widely studied. Lower intensity therapy (LIT) has been introduced for the treatment of older or so-called “unfit/frail” patients and, apart from the improved tolerability, it has resulted in longer overall survival compared to conventional care regimens [39]

The MFC-MRD-based prediction of prognosis in older patients with AML who are not eligible to receive intensive chemotherapy was investigated in many clinical trials [30]. Maiti et al. [31] analyzed the prognostic value of achieving negative MRD in this group of patients receiving first-line therapy, i.e., DEC10-VEN regimens. The achievement of MRD-negative status at 1 and 2 months after starting therapy was associated with better OS in older patients with AML with intermediate- and adverse-risk cytogenetics. MRD-negative status at 1, 2, and 4 months after starting therapy was associated with significantly better survival in older/unfit patients with AML.

The combination of venetoclax plus hypomethylating agents seems to be an effective treatment regimen leading to improvements in complete remission rates and overall survival. In a single center phase II study in patients with newly diagnosed (ND) AML older than 60 years of age, secondary AML (sAML), and relapsed or refractory (R/R) AML, the administration of azacytidine with oral venetoclax was highly efficacious and safe [40,41]. The value of MRD assessment was again evident as patients with composite complete remission with negative MFC-MRD based on the 0.1% cut-off had comparable 2-year OS regardless of treatment type (73.6% in the azacitidine–venetoclax group and 63.6% in the placebo group). In addition, for MRD-negative patients in the venetoclax arm, the median EFS and OS were not reached in patients with CR and MRD < 0.1%, whereas the ones with MRD > 0.1% had median EFS and OS at 10.6 and 18.7 months, respectively [32].

The PETHEMA-FLUGAZA phase III trial in elderly AML patients reported that MFC-MRD status was the only independent predictor of relapse-free survival in 72 patients who achieved CR either by the combination of low-dose cytarabine plus fludarabine or azacytidine at standard doses. However, even patients with MRD negativity exhibited substantial genetic abnormalities in CD34+ progenitors, indicative of the inability of the current MRD threshold to detect the existence of very low levels of clonal cells that can drive relapse [42].

### 2.3. MFC-MRD Prior to and after AlloSCT

In a meta-analysis of large-scale AML studies reported by Short et al., 2020, patients with MRD negativity had improved rates of overall survival (OS, 68% vs. 34%) and disease-free survival (DFS, 64%vs. 25%) at 5 years versus patients with MRD positivity [29]. Patients with MRD positivity may benefit from pre-transplantation strategies and MRD assessment may contribute to choosing the appropriate conditioning regimen for alloSCT [33,34]. Hourigan et al. demonstrated that reduced-intensity conditioning resulted in worse outcomes when compared to full conditioning in patients with MRD positivity [43]. Various studies showed that both pediatric and adult populations with AML who retain MRD positivity after induction treatment display a worse outcome following alloSCT compared to patients with MRD negativity. In a meta-analysis performed by Buckley et al., 2017, MRD positivity before alloSCT was associated with decreased leukemia-free survival (LFS, hazard ratio [HR] = 2.76), OS (HR = 2.36), and cumulative incidence of relapse (CIR, HR = 3.65) [44].

In the GIMEMA adult AML1310 trial [24], patients with intermediate-risk AML received either autologous-SCT or alloSCT depending on the level of FCM-MRD (threshold of 0.035%). Overall and disease-free survival (DFS) at 24 months was assessed, with values of 78.6% and 61.4% in MRD-positive and 69.8% and 66.6% in MRD-negative patients, respectively.

The National Cancer Research Institute’s AML17 trial is currently the largest study to investigate the key elements impacting MRD in AML. A total of 1874 adults < 60 years of age with AML were enrolled and treated with standard daunorubicin plus cytarabine based induction treatment followed by risk-adapted chemotherapy consolidation, with or without alloSCT. Patients with MRD positivity (defined as ≥0.1% by the MFC assay) after cycle 1 had similar 5-year overall survival as patients who only achieved a partial response (51% vs. 46%, respectively), emphasizing the poor outcomes associated with persistent MRD [11]. Other studies have also shown that MRD persistence prior to alloSCT has a negative predictive value, while MRD negativity in the same setting leads to favorable outcomes (3-year overall survival estimates of >70% and a relapse risk of 20–25%) after myeloablative alloSCT [35,45]. Interestingly, post-alloSCT MRD status appeared to be more informative than pre-alloSCT status in the study by Zhou et al. [36]. The 3-year overall survival rate for patients with pre-alloSCT MRD that persisted after the transplant was 19%, whereas in patients with pre-alloSCT MRD that cleared with transplantation it was 29%. Monitoring for residual disease post-alloSCT can be predictive of relapse and monthly monitoring during the first 6 months has been proposed for the early detection of relapse [27].

Of note, the combination of MFC-MRD with mutational analysis using NGS or digital PCR increased the sensitivity for the diagnosis significantly [28]. The first study to show that a targeted multigene NGS panel can be used to detect residual mutations in AML patients immediately before allogeneic transplantation was conducted by Getta et al. [23], who compared MFC-MRD with a multigene NGS assay for the first time and showed that the presence of residual mutations and atypical blasts is associated with post-transplantation relapse and survival. The burden of residual disease, as measured by residual leukemia alleles, was significantly higher than the percentage of aberrant blasts assessed by MFC, indicating the presence of residual leukemia alleles in non-blast compartments at the time of remission.

## 3. Future Perspective: A Holistic Approach for MFC-MRD in AML

As MRD is becoming an invaluable outcome predictor for AML, the need for standardization and improvement of sensitivity and specificity of the current methods is imperative. Crucial limitations still exist, as reflected by the fact that the false-negative rates of MFC-MRD in AML vary from 13–30% and that up to 70% of patients with low/negative MRD, based on the 0.1% cut-off, will finally relapse [46,47,48].

Beyond the technological and methodological limitations of MRD evaluation, there are other issues pertaining to the biological concept of MRD. Merely quantitative measurement of the residual leukemic cells, although established in several clinical settings [3,49], has obvious drawbacks. The individual properties of the remaining leukemic clone(s), namely the refractoriness and leukemogenic potential, are not fully addressed with the current panels of MFC-MRD, whereas persistent or donor-derived clonal hematopoiesis in the case of alloSCT is often difficult to distinguish from true residual AML [50]. Surface phenotypic characterization of leukemic subpopulations correlates poorly with the functional status of the latter and cannot dissect the wide inter- and intratumor heterogeneity of AML. Interrogation of intracellular signaling pathways, and molecular studies in xenograft models [51] revealed a decoupling of the surface phenotype of leukemic progenitors from their leukemia-regenerating capacity, indicating that MRD based solely on conventional surface phenotyping cannot reliably estimate the leukemogenic potential of the residual clonal cells.

An even more important issue is the fact that the mere measurement of residual leukemic burden does not take into account tumor–immune system interactions. Oncogenesis is not merely a cell-intrinsic process but involves complex interactions of the malignant clone with neighboring non-clonal cells [52]. The patient’s immune system can suppress but also promote cancerous growth and appears to sculpt tumor immunogenicity, as theorized by the cancer immunoediting model [53]. On the other hand, malignant neoplasms subvert the anti-tumor immune response by employing a myriad of mechanisms involving cellular and soluble immune elements both systemically and locally [54,55]. The ultimate consequence of this process is the development of resistant survivor clones that drive disease relapse after MRD “negativity”, as has been demonstrated in preclinical models [56,57]. The prognostic value of tumor-infiltrating T cell subsets is well established in solid tumors [58] but has not yet been incorporated in the risk assessment of AML, despite existing evidence for the prognostic relevance of the immune biosignature of AML [59,60].

The pivotal role of crosstalk between AML and immune cells in disease relapse after an initial remission is long known, but three recent studies provide intriguing data that arouse further concerns about the validity of conventional MFC-MRD. By analyzing primary AML samples, Christofer et al. demonstrated that relapse after allogeneic hematopoietic stem cell transplantation (alloSCT) is accompanied by a theoretically reversible epigenetic dysregulation of immune pathways, which may be important in immune evasion of the leukemic clone [61]. Almost identical findings were reported by Toffalori et al. who identified a predictive transcriptional signature in AML blasts of transplanted patients, characterized by upregulation of inhibitory ligands and subversion of antigen presentation [62]. Evidently, both biosignatures cannot be traced with current MRD methods and this may, in part, account for the currently unsatisfactory accuracy of MRD testing [63]. In addition to the above findings in leukemic blasts, a third study revealed the prognostic significance of cellular and soluble immune system elements by investigating the effect of immune reconstitution after alloSCT on the outcome of patients. Mass cytometry analysis of 89 immune cell subsets and serum profiling of soluble mediators were performed longitudinally after alloSCT and prognostically relevant immune signatures were constructed [64].

Though both targeted and conventional therapies can induce immune cell priming by several mechanisms, including immunogenic cell death and eradication of the immunosuppressive tumor milieu, the increasing use of low-intensity therapies in younger patients poses another challenge. In a phase II study combining azacytidine and nivolumab in relapsed/refractory (R/R) AML, the pretherapeutic size of the bone marrow T cell pool predicted response to treatment, whereas two CD4^+^ and CD8^+^ effector subsets expanded further post-treatment [65]. In addition, recent results from our group suggest that modulation of the IL−6/STAT3 signaling axis in conventional CD4^+^ T cells of high risk MDS patients is strongly associated with response and outcome to azacytidine, potentially representing immune-mediated antileukemic activity of the latter [66]. Similarly, venetoclax, the standard partner of azacytidine in the treatment of elderly and/or frail AML patients, has recently been shown to induce T cell antileukemic response by increasing the intracellular levels of reactive oxygen species in conventional and CD3^+^CD4^–^CD8^–^ double-negative T cells [67]. Collectively, these data suggest the pleiotropic action of newer agents by engaging a multitude of immune-mediated mechanisms and, at least partially, explain the frequently observed delayed responses and the dissociation between clinical response and overall survival [4,5].

Two studies of multiple myeloma reported that the concomitant measurement of immune cell subsets by flow cytometry considerably improves the prognostic power of MRD in diverse disease settings [68,69]. However, the complex tumor–immune cell interactions, the interindividual variability of immune systems, and the dynamic nature of immune reconstitution after AML treatment require more in-depth analysis with functional phenotyping and molecular characterization of immune components. On the other hand, a future application of a combined myeloid and immune-based MRD should be simple, standardized, and comprehensible in order to be used as a prognostic marker and/or endpoint in clinical trials. Designing an informative panel of immune markers for MFC-MRD in AML poses an obviously huge challenge considering the vast complexity of the immune system. However, based on the aforementioned studies, the immune panel can include markers of T helper and cytotoxic cell differentiation, i.e., FOXP3 and the intracellular cytokines interferon (IFN)-γ, IL-4, and IL-17 and TGFβ, IFNα/β, γ and λ receptors, immune inhibitory receptors, and their ligands (i.e., PD1, PDL1, TIM-3, CTLA-4) and cytolytic enzymes (perforin, granzymes A and B). Functional phenotyping can also be used in both T and blast cells in order to address ROS levels and the STAT signaling biosignatures (Figure 1).

## 4. Conclusions

Despite its unequivocal value in treatment guidance, MRD is not yet used routinely and has not been incorporated in the treatment algorithms relating to AML. Establishing MRD negativity as a surrogate endpoint for survival will accelerate drug selection and approval, whereas in everyday practice this will spare patients unnecessary and potentially harmful treatment approaches. However, the current methods of MRD assessment are solely focused on measuring the size of the residual leukemic clones without integrating the intrinsic genetic variability of the clonal cells and/or any metric of immune competence, thus overlooking the parallel alterations of the immune milieu (Figure 1). This fact may account for the still imperfect predictive accuracy of MRD as the status of antitumor immunity can critically affect disease progression, independent of the levels of residual leukemia. For instance, the spontaneous achievement of MRD negativity in post-chemotherapy MRD^+^ AML patients bearing the NPM1 mutation [70] potentially mirrors the gradual eradication of the residual leukemic burden through the restoration of an effective antileukemic immune response after chemotherapy. The simultaneous assessment of the alterations of the leukemic clone and the immune system has the potential to overcome the limitations of current MRD methods. A snapshot of the antitumor immune response at relevant post-treatment timepoints provides the possibility of quantifying the competence of antitumor immunity and redefining MRD status, independent of the persistence of measurable leukemic burden.

Flow cytometry remains the key tool for the comprehensive characterization of both myeloid and immune cells and its versatility is ideal for the synchronous study of these two pivotal players in the pathobiology of AML relapse. As flow cytometric technologies are rapidly advancing, and the combined analysis of immunophenotypic, spatial, and morphological traits is becoming a reality, the parallel study [71] of the kinetics of leukemic and immune components in AML could identify a serviceable and easily applicable MRD marker which would outcompete the current flow-based MRD method.

## Figures and Tables

**Figure 1 cancers-14-04294-f001:**
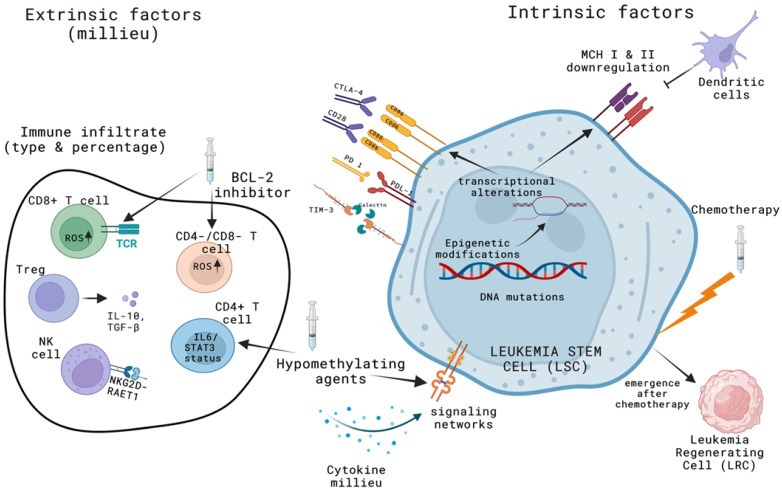
Quantifiable immune factors and other factors affecting leukemic relapse that can be used to increase MFC-MRD accuracy. (Created with BioRender.com, (Accessed on 8 August 2022)). Multiparametric flow cytometry can accurately assess post-treatment qualitative and quantitative alterations in immune and leukemic cell and bone marrow cytokines. Treatment with the hypomethylating agent azacytidine can modulate the signal transducer and activator of transcription (STAT) architecture in both CD4^+^ and CD34^+^ cells, while the bcl-2 inhibitor venetoclax can increase reactive oxygen species generation and boost the antileukemic activity of CD8^+^ and CD4/CD8 double-negative T cells. Conventional chemotherapy, on the other hand, leads to the emergence of a phenotypically and molecularly distinct subpopulation of leukemia-regenerating cells which cannot be traced with the current panels of MFC-MRD. Finally, relapse after allogeneic transplantation appears to be driven by epigenetic alterations in leukemic stem cells resulting in deregulation of the immune pathways involved in antigenic presentation and T cell co-stimulation.

**Table 1 cancers-14-04294-t001:** Advantages and disadvantages of methods used to detect MRD in AML.

Method	Sensitivity	Advantages	Disadvantages	References
Flow cytometry FC-LAIP (Leukemia-Associated Immunophenotypes)	10^−3^ to 10^−5^	Sensitivity Applicability to >90% of patients Rapid turnaround time Available technique through laboratories Can distinguish between live and dead cells	Experienced staff needed for proper interpretation Need for standardization Stability of the leukemic phenotype missing Diagnostic pretreatment sample needed Extended antibody panel needed Sensitivity depends on the antibody used	Brooimans 2019 [12] Maurer-Granofszky 2021 [13] [Wood 2016] [14]
Flow cytometry FC-DfN (Different from Normal)	10^−3^ to 10^−5^	Sensitivity Applicability to >90% of patients Diagnostic sample not required Phenotypic shifts do not affect the results Rapid turn-around time Can distinguish between live and dead cells	Need for standardization Experienced staff needed to operate the process, subjectivity in the definition of population Sensitivity depends on the antibody used	Schuurhuis 2018 [9] Maurer-Granofszky 2021 [13] Wood 2020 [15]
−19NGS	10^−3^ to 10^−5^	Limited applicability Easy to be conducted High sensitivity, theoretically to 10^−6^, depending on the NGS platform	Need for standardization Mutations can be identified in healthy populations (not necessarily linked with disease) Sample contamination Sensitivity is affected by error rate Clonal evolution (if based on allelic ratios)	Ngai 2021 [2] Dix 2020 [7]
RT-qPCR	10^−3^ to 10^−5^	High sensitivity (≥MFC) Quality assurance integration Applicability Standardization	Time-consuming Need for expertise Threshold limit settings required Expensive Sensitivity is affected as well by the expression level of the target per cell Molecular targets applicable to only ~50% of all AML patients (less in elderly)	Ngai 2021 [2] Wood 2016 [14]

**Table 2 cancers-14-04294-t002:** Studies with Multiparametric Flow Cytometry (MFC)-based MRD in AML.

Reference	No. of Patients	Age (Years) Median (Range)	Method	Cut-Off Level	Timepoint of MRD Assessment	Outcome
**Intensive Chemotherapy**						
Sievers et al., 2003 [20]	252 Pediatric	0–2 or 10–21	MFC-MRD	≥0.5% blasts	Before and after intensification therapy	Before MRDpos: Relative risk of relapse 4.2 (95% CI = 2.6–6.8, *p* < 0.0001) After MRDpos: 5.3 (95% CI = 3.2–8.6, *p* < 0.0001) All cohorts: Median time to relapse: MRDpos vs. MRD neg 168.5 days versus 293 days, *p* = 0.008) Relative risk of death MRDpos:3.4 (95% CI = 2.2–5.4, *p* < 0.0001)
Langebrake et al., 2006 [21] (2 Parts)	150	Part1 7.59 (0.2–17.7) Part 2 9.98 (0.06–20)	LAIP MFC-MRD	<0.1%	-BPM1 First at 15 days from the start of treatment -BPM2 Second at 29 days from the start of treatment	3-year EFS BPM1: ΜRDneg: 48% ± 9% ΜRDpos: 71% ± 6% *p* = 0.029 BPM2: ΜRDneg: 50% ± 7% ΜRDpos: 70% ± 6% *p* = 0.033
Fu-Jia Liu et al., 2021 [22]	492	45 (15–74)	MFC-MRD 10-color FC	0.1%	After induction therapy	MRDneg: <60, 276 (83.9%) ≥60, 53 (16.1%) MRDpos: <60, 136 (83.4%) ≥60, 27 (16.6%) *p* = 1.000
Getta et al., 2017 [23]	104	58 (21–78)	DfN MFC-MRD alone or combined with NGS 10-color MFC assay	0.1%	Pre-alloSCT	MFC–MRDneg: 18-month relapse 9% OS 73% MFC–MRDpos: 18-month relapse: 37% OS 48%
Vendittiet al., 2019 [24]	500	49 (18–60.9)	LAIP MFC-MRD combined with qPCR 8-color MFC assay	0.035%	After consolidation	Both neg: 2-year OS 89% and DFS 69% MFCpos/PCR neg or MFCneg/PCR pos: 2-year OS 88–89% DFS 65–76% Both pos: 2-year OS 55%, DFS 22%
Coustan-Smith et al., 2018 [25]	370	<1–63	Novel leukemia-specific markers	1 in 10^5^	At diagnosis	Clinical outcomes not determined
Terwijn et al., 2013 [26]	517	48 (18–60)	LAIPMFC-MRD	0.1%	After induction therapy	MRDneg: RFS > 47 months 4-year RFS 52% MRDpos: median RFS 8.6 months 4-year RFS 23%
Jacobsohn et al., 2018 [27]	144	Patients < 21 years of age	DfN MFC-MRD	0.02%	preHCT	MRDneg: 2-year relapse risk: 32% 2-year DFS: 55% 2-year OS: 63% MRDpos: 2-year relapse risk: 70% 2-year DFS: 10% 2-year OS: 20%
Daga et al., 2020 [28]	39 out of 41 patients	Adults > 60 years	Combination of MFC-MRD followed by (NGS) or digital PCR	0.1%	After induction therapy	MRDneg: 18 (48.2%) % relapse 27.8% MRDpos: 21 (53.8%) % relapse 71.4% *p* = 0.007 Median RFS 283 vs. not reached, *p* = 0.003 5-year CIR: 90.5% vs. 28%, *p* < 0.001 OS: not significant *p* = 0.085 Median follow-up time: 559 days
Short et al., 2020 [29] Meta-analysis of 81 studies	151	Adult-Pediatric	MFC-MRD, qPCR NGS, or cytogenetics/FISH	various	Induction or during/after consolidation	25 (40) MFC-MRD detection studies with OS analysis 29 (43) MFC-MRD detection studies with DFS analysis Overall analysis through all methods: MRDneg 5-y DFS: 64% 5-y OS: 68% MRDpos 5-y DFS: 25% 5-y OS: 34%
Wei et al., 2020 [30]	472	86 (55–86)	LAIP MFC-MRD	0.1%	First remission after IC	2-year survival MRDneg CC-486: 58.6% MRDpos CC-486: 39.5% MRDneg placebo: 51.7% MRDpos placebo: 22.0% 2-year survival differences (95% CI) MRDneg: 6.9 (5.8 to 19.5) MRDpos: 17.5 (5.3 to 29.8)
**Low-Intensity Chemotherapy**						
Mait et al., 2021 [31]	97 Venetoclax plus Decitabine	72 (68–78)	MFC-MRD	0.1%	1, 2, 4 months of therapy	MRDneg at 2 months: Median RFS, not reached vs. 5.2 months hazard ratio [HR] = 0.31; 95% CI, 0.12–0.78; *p* = 0.004 Median EFS, not reached vs. 5.8 months; HR, 0.25; 95% CI, 0.12–0.55; *p* = 0.001) MRDneg CR (median OS, 25.1 vs. 7.1 months; HR= 0.23; 95% CI, 0.110.51; *p* = 0.001) MRD neg at 1 month Median OS, 25.1 vs. 3.4 months; HR, 0.15; 95% CI, 0.03–0.64; *p* = 0.0001
Pratz et al., 2022 [32]	164 Azacitidine–Venetoclax (N = 286) Azacitidine–Placebo (N = 145)	76 (49–91) 76 (60–90)	MFC-MRD	0.1%	After cycle 1 and every 3 cycles	MRDneg Median EFS: not reached Median OS: not reached MRDpos Median EFS: 10.6 Median OS: 18.7
**Allo-HCT**						
Araki et al., 2016 [33]	359	50 (18.2–75.3)	DfN MFC-MRD 10 colors	0.1%	Pre-alloSCT	MRDneg: 3-year OS >70% Relapse risk 20–25% MRDpos: 3-year OS 25% Relapse risk 70%
Rubnitz et al., 2010 [34]	202	9.1 (2–21.4)	MFC-MRD	>0.1%	After induction I and II	After induction I: MRDneg 3-year CIR: 16.9% ± 3.4% 3-year EFS: 73.6% ± 5% MRDpos 3-year CIR: 38.6% ± 5.8% 3-year EFS: 43.1% ± 6.9% After induction II: MRD-neg 3-year CIR: 16.7% ± 3.1% 3-year EFS: 71.2% ± 4.7% MRDpos 3-year CIR: 56.3% ± 8.4% 3-year EFS: 35.8% ± 8.6%
Walter et al., 2011 [35]	99	45.3 (0.6–69.5)	DfN MFC-MRD 10-colors	0.1%	Before HCT	OS MRDneg (n = 75) HR:1 MRDpos (n = 24) HR: 4.05 95% CI = 1.90 to 8.62 *p* < 0.001 Relapse MRDneg HR:1 MRDpos HR: 8.49 95% CI: 3.67 to 19.65 *p* < 0.001 2-year OS MRDneg: 76.6% (64.4% to 85.1%) MRDpos: 30.2% (13.1% to 49.3%) 2-year DFS MRDneg: 74.8% (62.8% to 83.4%) MRDpos: 9.0% (1.6% to 24.9%)
Zhou Y et al., 2016 [36]	279	>18 years	DfN MFC-MRD 10 colors	Not used	Pre-alloSCT and post-alloSCT (day 28)	MRDpos pre-alloSCT and MRDneg post: 3-year OS: 29% 3-year RFS: 18% MRDpos at both timepoints: 3-year OS: 19% 3-year RFS: 14% MRDneg at both timepoints: 3-year OS: 76% 3-year RFS: 71

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
