# Peer review of "MRD Monitoring by Multiparametric Flow Cytometry in AML: Is It Time to Incorporate Immune Parameters?"

_cancers, 2022, doi:10.3390/cancers14174294_

Round 1
Reviewer 1 Report
Commendations:
- The authors have done well in proposing a novel approach to MFC-MRD assessments, namely to include the evaluation of leukemic clones in combination with metrics of anti-tumor immune response/immune competence in order to improve the predictive accuracy of MRD.
- The authors provided an extensive and thorough review of research articles and clinical trials that provide evidence in support of including MFC-MRD assessments as a standardized MRD technique that can be used as a surrogate endpoint for predicting patient survival. They also indicated that this can be very helpful in MRD-guided treatment regimens and clinical trials.
- The authors did an excellent job providing additional evidence for the use of MFC-MRD after the major treatment categories namely intensive chemotherapy, lower intensity treatment and alloSCT.
- The authors also highlighted additional approaches to the use of MFC-MRD by referencing the work of others who have utilized the combination of MFC-MRD with mutational analysis using NGS or digital PCR.
Suggestions:
- Since an important/major theme in the review article was that of incorporating the evaluation of ant-itumor immune responses in MFC-MRD assessments, then it may be best for the authors to review the title of their review article to reflect this. The current title does not fully capture the key messages that the authors are trying to convey.
- The intent for the question (found in the first sentence) in the introduction is not clear to readers since “remission” was not fully defined in the subsequent paragraphs. It is probably helpful for the authors to provide a definition so that readers who are not as knowledgeable about the field can also benefit.
- Figure 1 shows quantifiable intrinsic and extrinsic factors affecting leukemic relapse that can increase MFC-MRD. Many of these are immune related factors. Should the figure title include “immune and other factors” which is consistent with the theme of the paper?
- Since the authors have emphasized the inclusion of anti-tumor markers within MFC-MRD assessments, it would be helpful for them to assist other researchers who read the article by highlighting a list of markers (e.g. markers for immune cell subsets, checkpoint inhibitors, etc) that they suggest other researchers/clinicians can include in their MFC-MRD panel. The authors can use some of the articles they have referenced that included immune markers to assist with this.
Reviewer 2 Report
I would like to congratulate the authors for their efforts on a comprehensive discussion on multiparametric flow cytometry based MRD detection. Pessach et al have carefully reviewed the current practices in minimal/measurable residual disease monitoring in AML, in review titled: MRD Monitoring by Multiparametric Flow Cytometry in AML: Is it all about Measuring Blasts? Multiparametric flow cytometry remains the most efficient tool for AML diagnosis and it’s about time when a consensus in MFC based MRD is established. A combination of leukemia associated phenotype, immune effector markers and NGS may become the go to approach for MRD monitoring in future.
Minor comments :
1. Authors may consider combining flow based approaches in table and compare them to NGS and RT based approach. It depends on how they find it more readable. Also, not to exclude one major limitation of NGS based MRD is clonal evolution which can only be correctly estimated by NGS but based on allelic ratios, it can be uninformative.
2. The cut off labels in table 2 are slight confusing since they are written in both in European, where a decimal is noted as comma, and American numbering system where a decimal is written as a point. Please use point or dot to define decimal.
3. Please expand newly diagnosed (ND) AML and relapsed/refractory (R/R) AML when used first in the text around line 150, to make it easier for novice readers who I am sure will benefit from this review article.
